# Graphlet correlation distance to compare small graphs

**Jérôme Roux**[1]*, **Nicolas Bez**[2], **Paul Rochet**[3], **Rocío Joo**[4], **Stéphanie Mahévas**[1]

**1** UMR DECOD, IFREMER, BP 21105, Nantes Cedex, France, **2** MARBEC, IRD, Univ Montpellier, Ifremer, CNRS, INRAE, Sète, France, **3** ENAC, Toulouse, France, **4** Global Fishing Watch, Washington, DC, United States of America

* jerome.th.roux@hotmail.com

**Data Availability Statement:** The datasets generated during and/or analysed during the current study are available in the Jérôme ROUX GitLab repository, (https://gitlab.com/jerome-roux/project_small_graphs_comparison_v2).

## Abstract

Graph models are standard for representing mutual relationships between sets of entities. Often, graphs deal with a large number of entities with a small number of connections (e.g. social media relationships, infectious disease spread). The distances or similarities between such large graphs are known to be well established by the Graphlet Correlation Distance ($GCD$). This paper deals with small graphs (with potentially high densities of connections) that have been somewhat neglected in the literature but that concern important fora like sociology, ecology and fisheries, to mention some examples. First, based on numerical experiments, we study the conditions under which Erdős-Rényi, Fitness Scale-Free, Watts-Strogatz small-world and geometric graphs can be distinguished by a specific $GCD$ measure based on 11 orbits, the $GCD_{11}$. This is done with respect to the density and the order (i.e. the number of nodes) of the graphs when comparing graphs with the same and different orders. Second, we develop a randomization statistical test based on the $GCD_{11}$ to compare empirical graphs to the four possible null models used in this analysis and apply it to a fishing case study where graphs represent pairwise proximity between fishing vessels. The statistical test rules out independent pairing within the fleet studied which is a standard assumption in fisheries. It also illustrates the difficulty to identify similarities between real-world small graphs and graph models.

## Introduction

In ecology, the science of biological interactions, understanding the functioning of a group of individuals, be it a group of humans, animals, cells, etc, requires understanding the interactions between them [1]. For many years now, graphs and graph theory have been used to describe and study the organisation of groups of individuals [2, 3]. The simplest graphs allow for representing the presence of interactions within a group of individuals. The interactions are then, graphically, the edges between the nodes of the graph (one node is equal to one individual). Mathematically, a graph is formalised by an adjacency matrix [4], with a number of columns and rows equal to the number of individuals, and elements taking a value equal to 1 if there is an interaction between the individuals and 0 otherwise. While such binary graphs are

**Funding:** JR, Conseil Regional Pays de la Loire. SM, National research project TRACFLO, National Institute for Ocean Science. The funders had no role in study design, data collection and analysis, decision to publish, or preparation of the manuscript.

**Competing interests:** The authors have declared that no competing interests exist.

simplistic representations of relational structure, they can provide an essential and formal representation of various complex phenomena from diverse scientific fields such as protein-protein interaction [5] in biology or the interaction between social animals [6] in ecology. Comparing graphs can therefore allow us to compare groups with respect to the interactions they exhibit. There is abundant literature in graph theory aimed at comparing graphs [7–10].

This comparison is often done in a descriptive and qualitative way by comparing synthetic indicators of graph topology i.e. the configuration by which the individuals of a graph are connected [11]. For example, by comparing the distribution of the number of links that each individual has (i.e. degree distribution [12]) or the occurrences of certain motifs of links (i.e. subgraphs formed from a subset of nodes and edges [13]) between bundles of individuals (i.e. motif distribution [13]). These descriptive approaches were first performed in domains such as sociology [14], chemistry [15] and physics in the'90s, and more recently in neuroscience to compare brain graphs [16], in genomics to compare molecular graphs from different species [17] and in behavioral ecology [18–22] to compare the structure of social relationships.

The shift to quantitative graph comparisons with the introduction of similarity or distance measures [23] has resulted in the development of plenty of distances (see [9] for a recent review). Amongst these, the Graphlet Correlation Distance ($GCD$) was shown to not only outperform the others but also to be robust to order (i.e. number of nodes [24]) and density (i.e. ratio of the number of edges with respect to the maximum number of possible edges [25]) differences between the graphs compared [26, 27]. Graphlets are small connected induced subgraphs (formed from a subset of the vertices of the graph and all of the edges in that subset) [28, 29] of a graph defined up to isomorphism [30] (meaning that two isomorphic induced subgraphs are considered two occurrences of the same graphlet). They emerged as an accurate mining tool to provide topological information that is not exclusively local [31]. Graphlets generalise the degree distribution of a graph to the distribution of subgraphs connected to a node that is assigned a particular role (called orbit) [8, 32]. Yaveroğlu et al [27] showed that eleven orbits were sufficient to exhaustively describe a graph so that the topology of the graph can be summarised by the correlation matrix between these eleven vectors of orbits' degrees, called the Graphlet Correlation Matrix ($GCM$) [27]. The $GCD_{11}$ was thus defined as the Euclidean distance between the $GCM$ of the graphs [27].

To go beyond the comparison of simple descriptors of interactions between individuals, it is appealing to test functional hypotheses about these interactions [23]. One possible approach is to test whether a graph can be considered as an outcome of a specific random graph (null model). For example, Erdös-Rényi [33] is a graph model where the links between individuals are mutually independent. It can therefore be used as a null model to test the absence of correlation between the interactions of individuals. Some studies based on different graph comparison methods identified the similarities between empirical graphs and the outcomes of some random graph models [32, 34]. However, to the best of our knowledge, none of these approaches involved $GCD$ measures.

Most of the studies available in the literature focus on graphs with large numbers of nodes (several hundred or thousands) and very low edge densities ($\leq 0.1$) [35]. However, large graphs are not the only real-world graphs. In sociology, for example, the classical examples of Zachary's (1997) karate club network [36] and Sampson's (1968) monks' network [37] contain 34 and 18 nodes respectively. In ecology, food webs can be studied at the level of trophic groups rather than at the level of species or individuals [38] with a number of entities from 25 to 172. In fisheries, fleets may consist of only ten or a few dozen interacting fishing vessels [39]. Thus, there are multiple cases of small-size graph applications that deserve dedicated methodological development.

This work deals with two main gaps in the literature. First, we assess the ability of $GCD_{11}$ to correctly distinguish small simulated graphs from known model types (Erdős-Rényi [33], Fitness Scale-Free [40], Watts-Strogatz small-world [41] and geometric [42]) by a clustering approach [27, 43] using numerical experiments. We simulated graphs of different order and density, fluctuating the order from 10 to 100 to mimic the range encountered in some real small graphs [36–38], while the density range is completely covered from 0 to 1. We study the power of the $GCD_{11}$ to discriminate graphs with equal density and order and graphs that differ in density and order. Then, we perform a sensitivity analysis of the $GCD_{11}$ discrimination power to the values of density and order. This allows defining a domain of separability, i.e. an envelope of orders and densities for which the $GCD_{11}$ allows distinguishing between graphs. Second, we propose a statistical test based on the $GCD_{11}$ to evaluate whether an empirical graph can be considered as an outcome of a particular random graph. Finally, we illustrate the use of the $GCD_{11}$ to analyse small graphs characterising pairwise links between fishing vessels at sea. We test if these graphs can be outcomes of an Erdős-Rényi random graph model which would be indicative of vessels that cross independently at sea, an implicit assumption frequently made in fisheries.

## Materials and methods

### Graphlet Correlation Distance ($GCD_{11}$)

Yaveroğlu et al [27] recently proposed to compare graphs on the basis of the first eleven non-redundant orbits graphlets of up to four nodes. Considering a graph $G$ of order $N$, they first compute the $N \times 11$ matrix which contains for each node their orbits' degree i.e. the number of times the node is presented in each of the eleven orbits. The columns are called Graphlet Degree Distributions ($GDD$) [32] and the first column is the standard vector of degree values. Then, the Spearman's Correlation coefficient [44] is computed between all columns of the $GDD$ matrix to build an $11 \times 11$ matrix called the Graphlet Correlation Matrix ($GCM$). In this framework, the topology of a given graph $G$ is summarised by its Graphlet Correlation Matrix denoted $GCM_G$. The $GCD_{11}$ between two graphs $G_1$ and $G_2$ is then defined as the Euclidean distance between the upper triangular parts of their respective $GCM$:

$$GCD_{11}(G_1, G_2) = \sqrt{\sum_{i=1}^{11}\sum_{j=i+1}^{11}(GCM_{G_1}(i,j) - GCM_{G_2}(i,j))^2} \tag{1}$$

### $GCD_{11}$ on small synthetic graphs

The performance of the $GCD_{11}$ to identify similarities between small graphs is assessed with a numerical experiment using four different models of random graphs, namely the Erdős-Rényi ($ER$) [33], the Fitness Scale-Free ($SF$) [40], the Watts-Strogatz small-world ($SW$) [41] and the Geometric ($GO$) [42] models.

The Erdős-Rényi model is the simplest and most common uncorrelated random graph model. An Erdős-Rényi graph $ER(N, d)$ of order $N$ and edge density $d = 2m/(N(N-1))$ gets $m$ edges that are randomly and uniformly chosen among the $\binom{N}{2}$ possible edges [33]. This simple configuration results in an uncorrelated graph i.e., with a zero assortativity [45] meaning that there is no preferential attachment among nodes. In other words, the Erdős-Rényi random model generates graphs where edges are statistically independent of each other.

Fitness Scale-Free models $SF(N, d, \gamma)$ are derived [46] from Scale-Free Barbási-Albert models [47]. A graph is deemed scale-free when its node degree distribution $p(k)$ follows a power law $p(k) \sim k^{-\gamma}$ whose power $\gamma$ is generally observed empirically between 2 and 3 [48]. As a

compromise, we set $\gamma = 2.5$ in this study. In scale-free graphs, few nodes have high connectivity with respect to the average degree leading to the emergence of hubs as observed (or supposedly observed [49]) in some real-world graphs [50]. Fitness models are based on the attractiveness of each node [40]. When this follows a power law, the degree distribution also follows a power law as a scale-free graph [51].

The Watts-Strogatz small-world model was developed to address the lack of realism of the random graph model (Erdős-Rényi) by relying on the notion of "small-world" phenomenon [52] also known as six degrees of separation [53]. The Watts-Strogatz small-world model is an intermediate model between a regular graph [54] and an Erdős-Rényi graph allowing to reconcile local properties of one and global properties of the other [55]. A Watts-Strogatz small-world graph $SW(N, k, p)$ is generated from a ring lattice graph of order $N$, where each node is connected to its $2k$ nearest neighbours. Then, each edge is kept with probability $1 - p$ or reconnected to a randomly chosen node (without loop or multiple edges) with a probability $p$. The parameter $p$ thus controls the coexistence of "short-range" (from the ring lattice) and "long-range" (from the random reconnection) connections [55]. As the edge density is $d = \frac{2k}{N-1}$, it does not span all the possible values on the [0, 1] interval. In the numerical experiment below, we randomly added or removed some edges (between 10 and 50 depending on the order $N$) to reach the desired edge density when necessary. Some of the simulated graphs were thus no longer Watts-Strogatz small-world strictly speaking but remained highly similar to it. To bring out the small world property [56], we set $p = 0.05$.

The Geometric model allows to generate spatial graphs where the condition of existence of an edge between two nodes depends on their proximity [57]. A Geometric graph $GO(N, l, r)$ is generated by placing $N$ independent nodes uniformly at random in $\mathbb{R}^l$ and by connecting pairs whose distance is smaller than $r$ [42]. In this study, the distance threshold $r$ was chosen to obtain the desired density $d$. Geometric graphs are dominated by local interactions [58] leading to the emergence of community structure.

For each model $M \in \{ER, SF, SW, GO\}$ and for a given order $N$ and a given edge density $d$ we generate 100 graphs $G_M^i(N, d)$ with $i = 1, \ldots, 100$. For $M \in \{ER, SF, SW\}$ we use the R 4.1.3 [59] package igraph (v1.2.1) [60] to generate graphs (see Data Availability statement).

**Comparing graphs with the same order and edge density.** We define a set of 51 possible values of order $N$ describing two gradual incremental steps as $N \in \{10, 11, \ldots, 48, 49, 50, 55, 60, \ldots 90, 95, 100\}$ and a set of 101 possible values of edge density $d$ regularly spaced as $d \in \{0, 0.01, \ldots, 0.99, 1\}$. According to these two sets of order and density, for a given order $N$, and a given edge density $d$, for each couple of models $(M_1, M_2) \in \{ER, SF, SW, GO\}^2$ with $M_1 \neq M_2$, we computed all the pairwise $GCD_{11}$ between their 100 respective generated graphs to construct a $200 \times 200$ block symmetric distance matrix $D = \begin{bmatrix} D_{M_1, M_1} & D_{M_1, M_2} \\ D_{M_2, M_1} & D_{M_2, M_2} \end{bmatrix}$. The discriminating power of $GCD_{11}$ is assessed by the Area Under the Precision-Recall (AUPR) curve [43] computed as follows. For each distance threshold $\epsilon_k$, $k = 1, \ldots, 100$, regularly spanning the range of distance values of $D$, we compute:

- the true positives $TP$, as the number of distances between graphs from the same model smaller than $\epsilon_k$;

- the true negatives $TN$, as the number of distances between graphs from two different models greater or equal to $\epsilon_k$;

- the false negatives $FN$, as the number of distances between graphs from the same model greater or equal to $\epsilon_k$;

- and the false positives, *FP*, as the number of distances between graphs from two different models smaller than $\epsilon_k$.

Precision (P) and recall (R) are then defined as:

$$P(\epsilon) = \frac{TP(\epsilon)}{TP(\epsilon) + FP(\epsilon)} \tag{2}$$

$$R(\epsilon) = \frac{TP(\epsilon)}{TP(\epsilon) + FN(\epsilon)} \tag{3}$$

The diagonals of $D_{M_1,M_1}$ and $D_{M_2,M_2}$ are trivial (null distance between a graph and itself) and are thus excluded from these calculations. We also removed the diagonals of $D_{M_2,M_1}$ and $D_{M_1,M_2}$ when computing the area under the precision-recall curve to guarantee the same number of intra-model distances $D_{M_1,M_1}$ and $D_{M_2,M_2}$ and inter-model distances $D_{M_2,M_1}$ and $D_{M_1,M_2}$. The area under the curve of the precision given the recall curve (AUPR) is defined as:

$$AUPR_{M_1,M_2}(N, d) = \sum_{k=2}^{100} P(\epsilon_k)\Delta R(\epsilon_k) \tag{4}$$

where $\Delta R(\epsilon_k)$ is the change in recall from rank $k - 1$ to $k$. For each combination of order $N$ and edge density $d$, the resultant $AUPR_{M_1,M_2}(N, d)$ is used to fill an $|N| \times |d|$ matrix $A_{M_1,M_2}$. The discriminating power of the $GCD_{11}$ amongst the four tested models is set to the minimum value obtained over all possible comparisons. From the $\binom{4}{2}$ pairs of models, we thus finally fill a $|N| \times |d|$ matrix $A$ as:

$$A(N, d) = min_{i \neq j}(A_{M_i,M_j}(N, d)) \tag{5}$$

An AUPR score of 1 means a perfect distinction between $M_1$ and $M_2$ (i.e. two clusters without overlapping) whereas an AUPR score of 0.5 is the expected score of a random classifier. An AUPR score of 0 occurs when graph topologies are all identical. We arbitrarily consider that an AUPR larger than 0.9 allows discrimination between two models amongst the four that are used and defined the domain of separability as the set of orders and densities providing an AUPR larger than 0.9. In the domain of separability, the $GCD_{11}$ is small (resp. large) between graphs coming from the same (resp. different) models amongst the four models used in the study. In other words, the domain of separability reflects the ability of the $GCD_{11}$ to discriminate graphs of two different models into two weakly or non-overlapping clusters.

**Comparing graphs with different orders and edge densities.** For each couple $(M_1, M_2)$ $\in \{ER, SF, SW, GO\}^2$ with $M_1 \neq M_2$, and for all possible pairs of combinations of orders and densities $(N, d) \times (N', d')$ we also build the three following $100 \times 100$ $GCD_{11}$ matrices using the already simulated graphs:

$$D_{M_1(N,d),M_1(N',d')}[i,j] = GCD_{11}(G^i_{M_1}(N, d), G^j_{M_1}(N', d')) \tag{6}$$

$$D_{M_2(N,d),M_2(N',d')}[i,j] = GCD_{11}(G^i_{M_2}(N, d), G^j_{M_2}(N', d')) \tag{7}$$

$$D_{M_1(N,d),M_2(N',d')}[i,j] = GCD_{11}(G^i_{M_1}(N, d), G^j_{M_2}(N', d')) \tag{8}$$

for $i, j = 1, \ldots, 100$. We then computed the proportion of cases where the inter-model distance $D_{M_1(N,d),M_2(N',d')}$ was larger than the two intra-model distances $D_{M_1(N,d),M_1(N',d')}$ and

$D_{M_2(N,d), M_2(N',d')}$. We consider this proportion as the probability that the $GCD_{11}$ assigns a smaller distance between two graphs from the same model than between two graphs from different models. A probability of 1 means that all intra-model distances are smaller than all inter-model distances. This probability is used to fill an $(|N| \times |d|) \times (|N'| \times |d'|)$ asymmetric matrix of probability $B_{M_1,M_2}((N,d),(N',d'))$. As in the previous section, all the possible comparisons were combined into a $(|N| \times |d|) \times (|N'| \times |d'|)$ synthetic matrix of probability as:

$$B((N,d),(N',d')) = min_{k \neq l}(B_{M_k,M_l}((N,d),(N',d'))) \tag{9}$$

We arbitrarily consider that a probability larger than 0.9 allows discrimination between two models amongst the four that are used and allows us to define the domain of separability. We also defined the surface of the domain of separability as the proportion of cells in the matrix of probability where the value is larger than 0.9. To limit the computing time and because the outputs change slowly with the order values, the number of possible values for the order and density are reduced so that $(N,d) \in \{10, 20, \ldots, 100\} \times \{0, 0.02, \ldots, 1\}$.

To visualize distances between graphs we use a principal component analysis (PCA) [61] where each observation (graph) is described by 55 variables (the 55 graphlet correlation coefficients in the $GCM$). Since PCA is based on the Euclidean distance as the $GCD_{11}$, to preserve the properties of the $GCD$, the PCA is computed on the 55 centered variables without scaling.

## Statistical test

In order to test if an empirical graph $G(N,d)$ is an outcome of an $M(N,d)$ random graph model ($H_0$) with $M \in \{ER, SF, SW, GO\}$, we build the following randomization statistical test. First, we simulate 1000 independent outcomes $M_k$ ($k = 1, \ldots, K = 1000$) of each possible reference model $M$. Second, we compute their Graphlet Correlation Matrices $GCM(M_k)$ and their average:

$$\overline{GCM}_M = \frac{1}{K} \sum_{k=1}^{K} GCM(M_k) \tag{10}$$

where $\overline{GCM}_M$ denotes the average Graphlet Correlation Matrix of $M$. Third, we compute the Graphlet Correlation Distance $\delta_{M_k}$ between $GCM(M_k)$ and $\overline{GCM}_M$ and the Graphlet Correlation Distance $\delta_G$ between $GCM_G$ and $\overline{GCM}_M$:

$$\delta_{M_k} = \sqrt{\sum_{i=1}^{11} \sum_{j=i+1}^{11} \left( \overline{GCM}_M(i,j) - GCM(M_k)(i,j) \right)^2} \tag{11}$$

$$\delta_G = \sqrt{\sum_{i=1}^{11} \sum_{j=i+1}^{11} \left( \overline{GCM}_M(i,j) - GCM_G(i,j) \right)^2} \tag{12}$$

Under $H_0$, $P(\delta_G < \delta) = P(\delta_{M_k} < \delta)$ with $\delta \in \mathbb{R}^+$, and the $p$-value for testing $H_0$ is calculated as $P(\delta_{M_k} > \delta_G)$. We computed $\eta$ as the number of times the distance $\delta_G$ between $GCM_G$ and $\overline{GCM}_M$ is smaller or equal than the distance $\delta_{M_k}$. The $p$-value is then defined by $\hat{p} = (\eta + 1)/(K + 1)$ [62]; the larger the $p$-value, the less evidence against $H_0$. To account for the difference in variability between the correlation coefficients of each pair of orbits, we also investigated the use of a standardised distance which provided very similar outcomes (S1 Eq).

## Empirical graphs

The developments proposed in this paper are illustrated on small graphs based on fisheries data. Joo et al [39] identified pairwise relationships between vessels of some fishing fleets (groups of vessels sharing the same technical characteristics) based on joint-movement analysis [63] of their GPS-tracks at sea. On the basis of this previous work, we derive a set of twenty graphs describing the relationships (the edges) between a set of trips (the nodes) within a fleet of French trawlers with unknown topological properties and of unknown types.

# Results and discussion

## Same orders and densities

The domain of separability between different models amongst {*ER*, *SF*, *SW*, *GO*} is globally parabolic with regards to the order and the density. The range of edge densities allowing clear discrimination increases with the graphs' order. For instance, when comparing graphs coming from Erdős-Rényi (*ER*) and Fitness Scale-Free (*SF*) models (Fig 1a), for orders of 25 and 50, the domain of separability respectively spans a range of edge densities approximately from 0.3 to 0.6 and from 0.1 to 0.95, respectively. Furthermore, a perfect discrimination (AUPR = 1) is gradually reached for graphs with more than 50 nodes, more and more irrespective of the edge density.

The parabolic shapes of the surfaces of the domain of separability described by the AUPR isolines are qualitatively similar but not identical regardless of which pair of models are compared (S1a–S1e Fig), with the exception of the Erdős-Rényi (*ER*) vs. the Geometric (*GO*) model (S1a Fig), where the domain of separability exhibits a strong asymmetrical surface with regards to the density.

The combination of all domains of separability (Fig 1b) appears as a mixture of all the domains of separability of each pair of models. The asymmetrical domain of separability between Erdős-Rényi and Geometric model (S1a Fig) is easily recognisable between the densities from 0.1 to 0.2, as well as the one between Erdős-Rényi and Watts-Strogatz small-world model (S1b Fig) for densities from 0.7 to 0.9.

The effect of the order and the density on the performance of the $GCD_{11}$ are related to the response of the different graphlet correlation coefficients to changes in order and density (Fig 2).

For a given density, the variability of each graphlet correlation coefficient is very high for small orders (lower triangle on Fig 2) leading to a small difference between the graphlet correlation coefficients of the four different models which are strongly overlapping. With increasing order (upper triangle on Fig 2), the variability of the graphlet correlation coefficients decreases leading to better discrimination (the domain of separability) between the four different models notably for some pairs of orbits in ramified graphlets of order 4 from $O_6$ to $O_{11}$, (except $O_7 \times O_{11}$). In other words, the increase in order allows to stabilise the graphlet correlation coefficients. Two reasons could explain this phenomenon. On one hand, the increase in order allows the emergence of complex topologies signing the topological properties of the model [64]. And finally, Spearman's Correlation coefficient becomes more accurate when computed on a larger number of nodes [65].

For a given pair of orbits, the "starting point" from an empty graph (density $d = 0$) and the "ending point" at a complete graph (density $d = 1$) are at the same values of correlation coefficient regardless the graph model (isomorphic graphs [30], black crosses Fig 1). The graphlet correlation coefficients are thus strongly similar amongst models when the density approaches those two extremes.

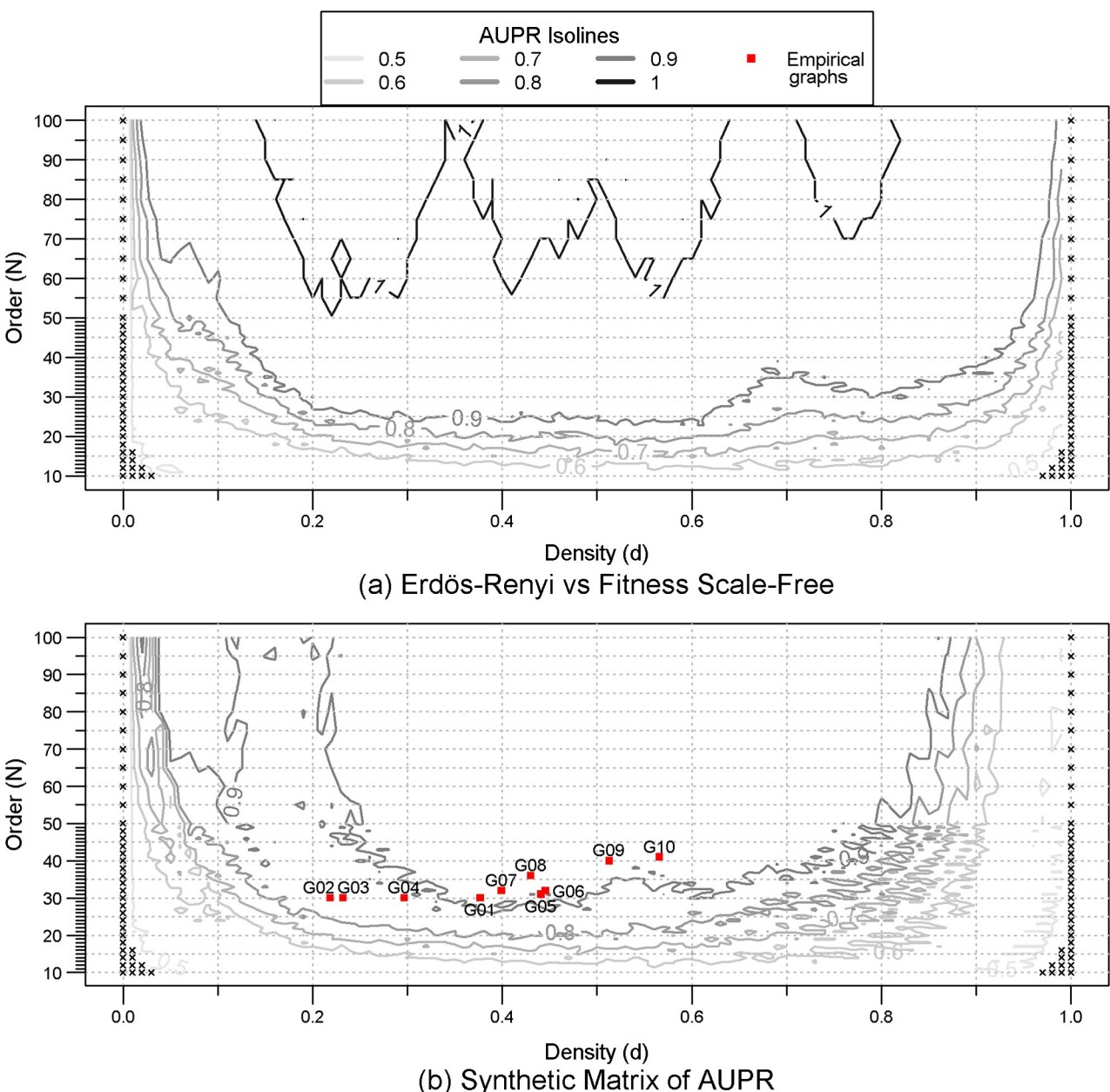

**Fig 1. Quality of the discrimination between models (AUPR).** (a) Diagram of the matrix of AUPR Erdős-Rényi vs Fitness Scale-Free ($A_{ER, SF}$), (b) Diagram of the synthetic matrix of AUPR ($A(N, d)$). For each pair of models, and for each order (from 10 to 100) and edge density (from 0 to 1) combination, the quality of clustering between 100 graphs of each of the two types of models is assessed by the Area Under the Precision-Recall curve (AUPR). A maximum value of 1 corresponds to perfect discrimination. Black crosses represent zero AUPR. Empirical graphs (red squares) are projected according to their features (order and edge density).

## Different orders and densities

When dealing with different orders and densities, the domain of separability of the $GCD_{11}$ turns out to depend first on the order. For instance, when comparing graphs coming from Erdős-Rényi (*ER*) and Fitness Scale-Free (*SF*) models (Fig 3a), for equal orders (Fig 3b, block diagrams on the first bisector), the surface of the domain of separability increases from 0.045

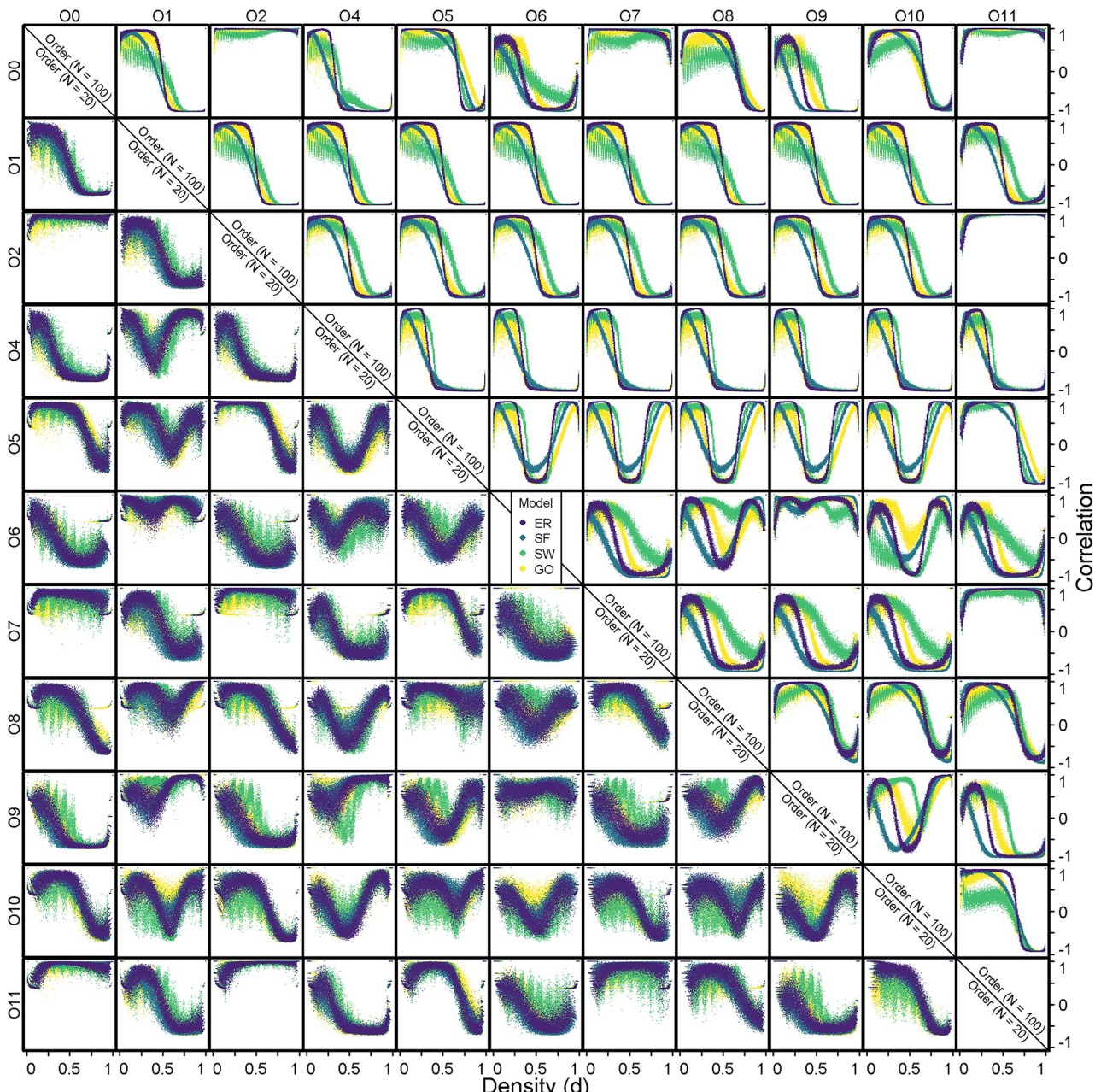

**Fig 2. Evolution of the 55 graphlet correlation coefficients.** Erdős-Rényi (*ER*, in dark purple), Fitness Scale-Free (*SF*, in blue), Watts-Strogatz small-world (*SW*, in green) and Geometric (*GO*, in yellow) models. For two different order values $N = 20$ (lower triangle figures) and $N = 100$ (upper triangle figures), the graphlet correlation coefficients are computed for 100 graphs of the four models and for edges densities ranging from 0 to 1.

to 0.19 when the order increases from 30 to 100. As with the same orders and densities, this means that due to reduced variability of the graphlet correlation coefficients, the edge density difference allowing clear discrimination between *ER* and *SF* is larger for "large" graphs. However, even for graphs with the same order, the difference in edge density allowing clear discrimination remains limited (Fig 3a).

This can be further detailed by principal component analyses (PCA) computed for a few cases taken as examples. Erdős-Rényi and Fitness Scale-Free graphs of order $N = 100$ and

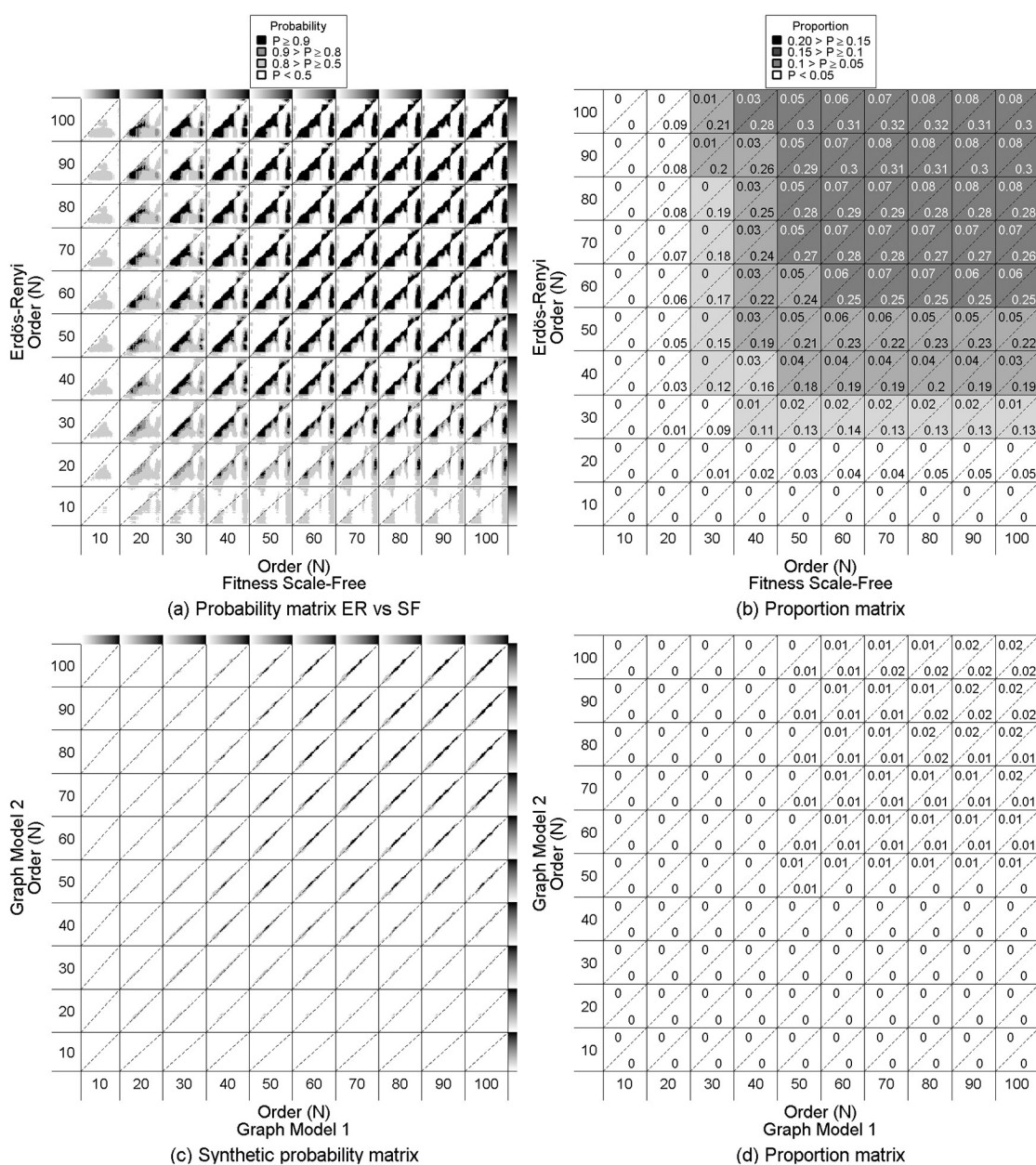

**Fig 3. Probability of correctly distinguishing two random graph models with different order and/or edge density.** Each block (*i*, *j*) concerns the comparison of an $M_1$ of order $N$ and a $M_2$ of order $N'$, with edge density $d$ and $d'$ respectively ranging from 0 to 1 (grey gradient from white to black on the top and right side) with $(M_1, M_2) \in \{ER, SF, SW, GO\}^2$ and $M_1 \neq M_2$. Dashed lines in each block highlight comparison when $d = d'$. (a) Probability matrix $B_{M_1, M_2}$ that $D_{M_1(N,d), M_2(N',d')} > max(D_{M_1(N,d), M_1(N',d')}, D_{M_2(N,d), M_2(N',d')})$ with $M_1 = SF$ and $M_2 = ER$. (b) Proportion of cells with a probability $P \geq 0.9$ in the triangle under or above the diagonal (cells covered by diagonals are not counted). Their mean quantifies the surface of the domain of separability of the $GCD_{11}$. (c) Synthetic matrix of probability matrix $B((N, d), (N', d'))$ according to each pair tested $(M_1, M_2) \in \{ER, SF, SW, GO\}^2$. (d) Proportion of cells in the synthetic matrix of probability with a probability $P \geq 0.9$.

density $d = 0.2$ are compared with similar graphs of order $N = 100$ and densities in $d' = \{0.4, 0.6, 0.8\}$. In each case, we consider the matrix $400 \times 55$ whose lines are the 55 graphlet correlation coefficients (upper or lower triangle of the *GCM*) for each of the 100 simulated graphs of each type and of each density. Over the three cases considered, the first two principal

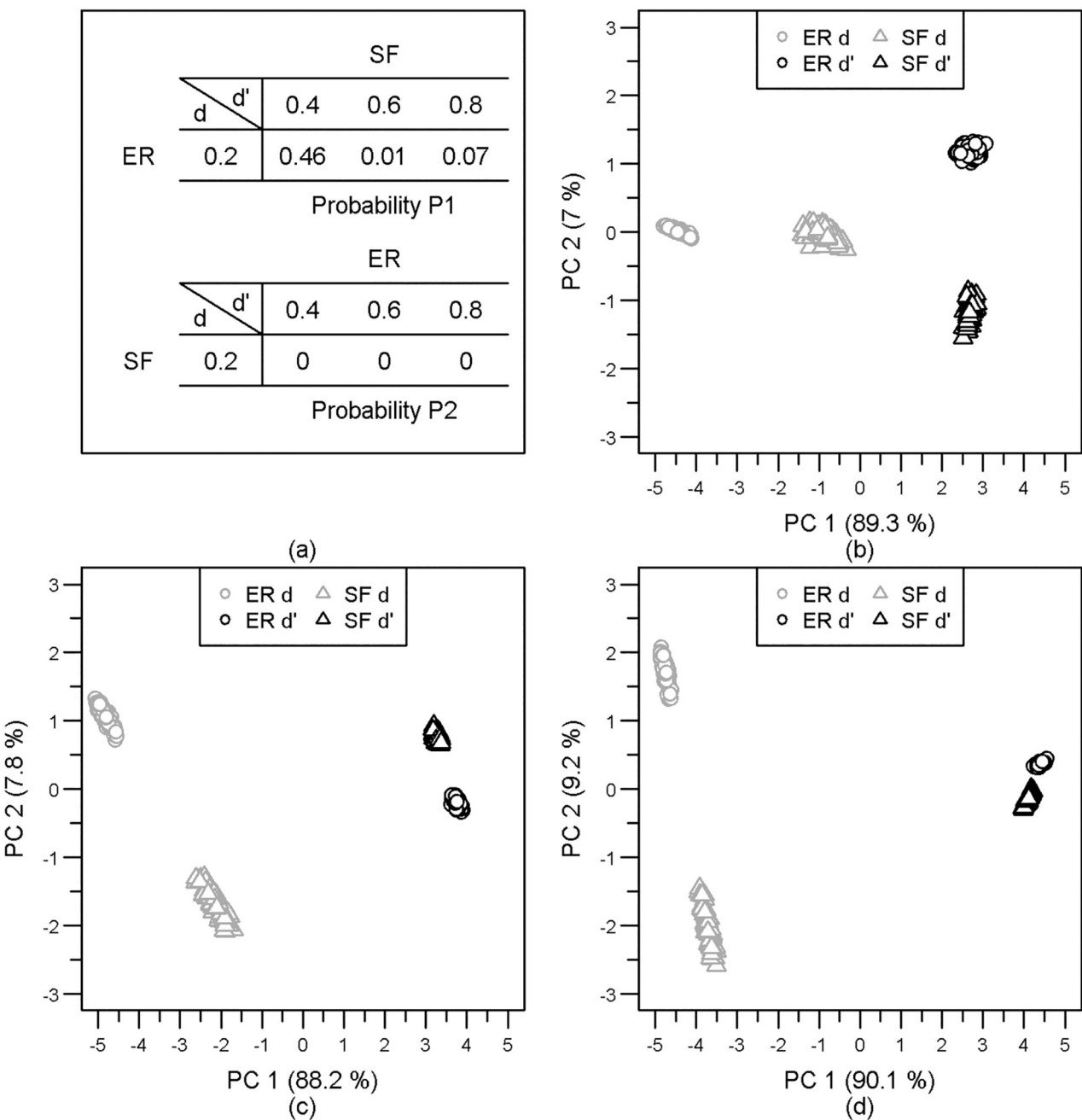

**Fig 4. PCA between Erdős-Rényi and Fitness Scale-Free graphs of order $N = 100$ and different edge density.** (b) ($d = 0.2$, $d' = 0.4$), (c) ($d = 0.2$, $d' = 0.6$), (d) ($d = 0.2$ $d' = 0.8$). For each density, 100 graphs of each model are projected according to their 55 graphlet correlation coefficients (upper or lower triangle of the *GCM*). (a) The two probabilities $P$ associated with each comparison in the matrix of probability $B_{ER, SF}$ are respectively $P1$ for $B_{ER\ (100,\ d),\ SF(100,\ d')}$ and $P2$ for $B_{ER(100,\ d'),\ SF(100,\ d)}$. With increasing edge density differences (b),(c) and (d) the two groups of dense graphs gradually become more and more similar.

components explain at least 96% of the input variability (Fig 4). There are two orthogonal linear combinations of the graphlet correlation coefficients that represent the same amount of information as the 55 coefficients. Given that the 2D space of the two principal components contains nearly 100% of the initial variability, the euclidean distance in this 2D space is a

suitable proxy for the $GCD_{11}$. Even a small difference in density (Fig 4b) leads to the division of the graphs into four groups, discriminating first the model and secondly the density. As the density discrepancy increases (Fig 4c and 4d), the denser graphs of the two different models are closer than the less dense ones. This increasing similarity is explained by the growing similarity between the graphlets correlation coefficients with the density (Fig 2). The distances between the graphs associated with two different densities for a given model are larger than those between the two different models. In other words, the $GCD_{11}$ does not discriminate graphs from different models with different densities because their topological differences do not depart more than the topologies of the same graph for different densities.

Let us consider the reference comparison cases where the two graphs are of the same order (block diagrams on the first bisector in Fig 3b). Increasing the order for one of the two models by ten leads systematically to larger domains of separability and this increase is amplified when increasing the order of the *ER* model. For instance, starting with the comparison between $ER(30, .)$ and $SF(30, .)$ the domain of separability equal to 0.045, then, the domain of separability expands from 0.045 to 0.11 when the order of the *ER* graph increases (in columns), while it flattens around 0.07 when the increase of order concerns the *SF* graph (in rows).

The domain of separability is also systematically asymmetric favouring situations where the edge density of the *SF* graph is larger than the edge density of the *ER* graph it is compared to, whatever their respective orders (Fig 3a). The exhibited asymmetry of the domain is, however, dependent on the edge densities. As a matter of fact, when the orders increase, the domain of separability acquires a "violin" shape consisting of a body for the lower half range of edge density, a head for high or very high density, and a neck that appears as a transition between the body and the head.

While the comparison between the *ER* and *SF* model produces a domain of separability with a violin shape, the other model comparisons produce domains of separability with other shapes (S2 Fig) and surface. The largest separability domain is obtained when comparing Fitness Scale-Free and Watt-Strogratz small-world models (S2c Fig) with a strong asymmetrical shape favouring a situation where the edge density of the *SF* graph is larger than that of the *SW*. These results differ from those of the same order and density (Fig 1 and S1 Fig) where the diagrams of different pairs of models were quite similar. Due to this strong variability, the synthetic domain of separability $B((N, d), (N', d'))$ (Fig 3c) obtained by crossing all of the two-by-two probability matrices $B_{M_1,M_2}$ is symmetric, very small and concentrated around the same density comparisons. Based on these results, the use of the $GCD_{11}$ seems relevant only when the densities are very close. Furthermore, at least an order of $N = 50$ is required to obtain a domain of separability.

## Empirical graphs comparison

**Empirical graphs features.** The empirical graphs used in this study are characterised by small orders ranging from 30 to 41 nodes and large edge densities ranging from 0.22 to 0.57 (Table 1). These empirical graphs (Fig 5a) show multiple dense components as a graph of communities. Despite the short observation time (1 week), these graphs are sensitive to finer-scale temporal variations with some trips occurring at the beginning of the week and others at the end prohibiting an encounter between them.

The *GCM* of empirical graphs (Fig 5b) exhibits a standard shape [27] with strong positive and negative correlations between the first eleven non-redundant orbits. These contrasted correlations capture heterogeneity in the role of trips (nodes) in the graph. For instance, the negative correlations between orbits {4, 6, 9} and orbits {0, 2, 5, 7, 8, 10, 11} indicates the existence of peripheral nodes [27] such as nodes {1, 6, 8, 14} which could reflect the solitary behaviour of

**Table 1. Main features of empirical graphs: Order (number of nodes), size (number of edges), and edge density (ratio between the size and the graph maximum size).**

| Graph | Order (*N*) | Size (*S*) | Density (*d*) |
|---|---|---|---|
| Graph 01 | 30 | 164 | 0.38 |
| Graph 02 | 30 | 101 | 0.23 |
| Graph 03 | 30 | 95 | 0.22 |
| Graph 04 | 30 | 129 | 0.3 |
| Graph 05 | 31 | 205 | 0.44 |
| Graph 06 | 32 | 221 | 0.45 |
| Graph 07 | 32 | 198 | 0.4 |
| Graph 08 | 36 | 271 | 0.43 |
| Graph 09 | 40 | 400 | 0.51 |
| Graph 10 | 41 | 464 | 0.57 |
| Mean | 33.2 | 224.8 | 0.39 |
| Range | [30; 41] | [95; 464] | [0.22; 0.57] |

some vessels. Conversely, the strong positive correlations between orbits {0, 2, 5, 7} reflect the existence of a hub such as node 20 [27] which could reflect a strong sociability behaviour.

**Testing model type.** Due to their small orders, some of the empirical graphs (Graph {02;03;04}, red squares in Fig 1b) are out of the synthetic domain of separability. However, being outside the domain of separability does not preclude the use of the test. Being outside of the domain of separability means that it is difficult to systematically separate graphs coming from different models. However, the rejection of the null hypothesis, which reflects the strong dissimilarity between the empirical graph and the null model, remains valid.

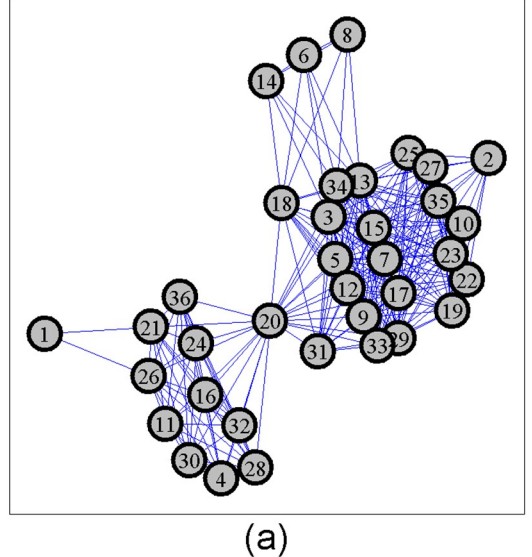

(a)

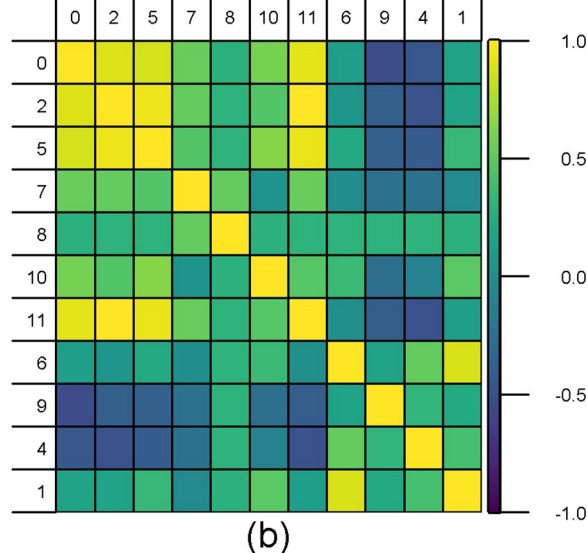

(b)

**Fig 5. Illustration of an empirical graph and its Graphlet Correlation Matrix.** (a) Example of an empirical graph (Graph 08) and (b) its Graphlet Correlation Matrix (*GCM*). The 11 non-redundant orbits are grouped according to their *role*, orbit {0} represents the familiar degree, {2, 5, 7} represent node in chain, {8, 10, 11} represent node in cycle, and {6, 9, 4, 1} represent terminal node. Cell colours correspond to the value of the correlation coefficient between the 11 non-redundant orbits from 1 (yellow) to −1 (dark purple).

**Table 2. Estimated p-values.** Each empirical graph is associated with an estimated *p*-value ($\hat{p}$) of being an outcome of an Erdős-Rényi, Fitness scale-free model, a Watts-Strogatz small word or a Geometric model. As in Table 1, empirical graphs are sorted according to their order.

| Graph | p-value | | | |
|---|---|---|---|---|
| | **Erdős-Rényi** | **Scale-Free** | **Watts-Strogatz** | **Geometric** |
| Graph 01 | 0.001*** | 0.001*** | 0.001*** | 0.051 |
| Graph 02 | 0.002** | 0.001*** | 0.002** | 0.122 |
| Graph 03 | 0.001*** | 0.001*** | 0.001*** | 0.088 |
| Graph 04 | 0.008** | 0.032* | 0.001*** | 0.029* |
| Graph 05 | 0.001*** | 0.001*** | 0.001*** | 0.154 |
| Graph 06 | 0.001*** | 0.001*** | 0.001*** | 0.002** |
| Graph 07 | 0.001*** | 0.001*** | 0.001*** | 0.002** |
| Graph 08 | 0.001*** | 0.001*** | 0.001*** | 0.009** |
| Graph 09 | 0.001*** | 0.001*** | 0.001*** | 0.011* |
| Graph 10 | 0.001*** | 0.001*** | 0.001*** | 0.337 |

($\hat{p}^{*} < 0.05$, $\hat{p}^{**} < 0.01$ and $\hat{p}^{***} \leq 0.001$).

According to our statistical test, none of the empirical graphs present any similarity with the same order and density graphs coming from Erdős-Rényi, Fitness Scale-Free and Watt-Strogratz small-world models (Table 2).

In terms of graphlet correlation coefficients, while there are similarities between the empirical *GCM* and the *GCMs* of the null model, for instance, Erdős-Rényi (Fig 6b), some empirical graphlet correlations are very different from those of the Erdős-Rényi. Interestingly these strong differences occur in orbits of high-order graphlets and specifically in the pair of orbits which contain the orbit $O_{10}$, for instance, $(O_9, O_{10})$, $(O_4, O_{10})$, $(O_1, O_{10})$, $(O_2, O_{10})$, or $(O_{10}, O_{11})$. A deep investigation of the role of orbit $O_{10}$ could help to understand the topological differences of these graphs.

The comparison between the empirical graphs and graphs coming from the Geometric model leads to more contrasting results. While empirical graphs {04, 06, 07, 08, 09} are significantly not similar to Geometric graphs, the test does not reject that empirical graphs {01, 02, 03, 05, 10} can be considered as outcomes of a Geometric graph mode. However, empirical graphs 02 and 03 are out of the domain of separability which does not allow us to conclude that these graphs are geometric graphs. For empirical graphs 01 and 05 which are in the domain of separability, the small *p*-values (0.051 and 0.154), one being just above the threshold for rejection, can be interpreted in relation to the definition of the domain of separability. Indeed, the AUPR larger than 0.9 associated with features of Graphs 01 and 05 (Fig 1b) implies a small, however not null, overlap between graphs coming from different models in {*ER*, *SF*, *SW*, *GO*}. This does not exclude the existence of "extreme" graphs from these models which might present some similarities. The largest *p*-value associated with Graph 10 suggests a strong similarity with Geometric graphs. Indeed, there is a strong similarity between most of the graphlet correlation coefficients of the empirical *GCM* of the graph 10 and the *GCMs* of Geometric graphs (Fig 6a). This suggests that this empirical graph and Geometric graphs share similar topological properties.

To account for the contrasting variabilities of the correlation coefficients of each pair of orbits (boxplots in Fig 6), we also built a statistical test based on the standardised distance between $GCM(M_k)$ and $\overline{GCM}_M$ (S1 Eq). This second test leads to similar conclusions (S1 Table) but rejects more clearly the null hypothesis. These results suggest that the standardised distance provides a more stringent test.

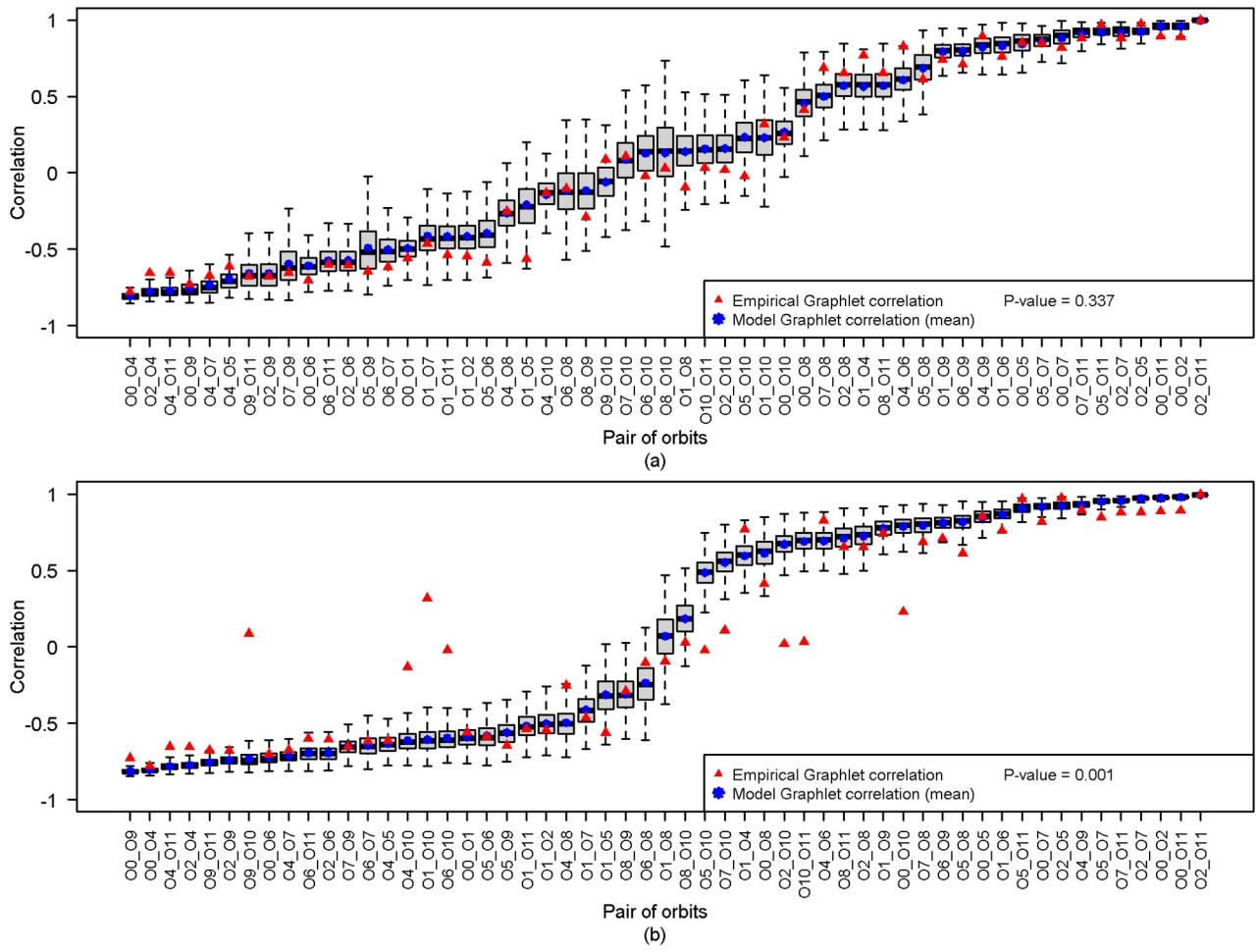

**Fig 6. Comparison between graphlet correlation coefficient of empirical and random graph model.** (a) Graph 10 and Geometric model (b) Graph 10 and Erdős-Rényi model. For each pair of orbits, the graphlet correlation coefficients of 1000 Erdős-Rényi or Geometric graphs are presented as a boxplot with the mean value (blue asterisk) and the empirical value (red triangle). The correlation coefficients are mostly different from Erdős-Rényi graphlet correlation coefficients. Conversely, there is a strong similarity between graphlet correlation coefficients from Geometric graphs.

## Conclusion

This work extends the use of the graphlet correlation distance ($GCD_{11}$) originally proposed for large real-world graphs to small real-world graphs. Through a numerical benchmark study based on four contrasted and commonly encountered random graph models, the Erdős-Rényi, Fitness scale-free, the Watts-Strogatz small world, and the Geometric graph models, we define the order × density domain within which the Graphlet Correlation Distance ($GCD_{11}$) clearly separate graphs with the same order and the same density configuration. While the performance of the $GCD_{11}$ is convincing to compare small graphs with the same order and the same density, when the orders and/or the densities differ, the performance of the $GCD_{11}$ deteriorates quickly and exhibits a high variability depending on the pair of models compared. Therefore, it seems essential to systematically check the applicability of the $GCD_{11}$ before comparing graphs of a different order and/or density to ensure the relevance and interpretability of its results.

For the four models used in this study, we show that the behaviours of the graphlets correlation coefficient in response to a change in density remain qualitatively similar. Furthermore,

the proposed experimental design and numerical analysis can be directly used with other random graph models to explore new properties of the $GCD_{11}$ and deeper investigate its domain of separability or to investigate other orbit combinations.

The statistical test and its standardised version proposed in this study to test the similarity between empirical graphs and graph models regardless of order and edge density can be applied without restriction on the size of the graphs. Some limitations of the $GCD_{11}$ are highlighted on the basis of numerical evidence presented here.

The application to fisheries data originally intends to test whether certain fishing behaviours can be considered independent. This property is generally required to apply statistical inference methods and more particularly when estimating population biomass in marine ecosystems. However, our results suggest that the graphs issued from the GPS tracks are not similar to Erdős-Rényi graphs. Some of them are rather Geometric.

Finally, in the context of an application of the $GCD$ on small graphs, the present work is also an opportunity to question the relevance of the choice of the 11 selected orbits for the construction of the $GCM$. Indeed, we have shown that there is a strong redundancy of the information provided by the different coefficients of graphlet correlations between pairs of orbits. Moreover, it seems that the information carried out by the correlation matrices ($GCM$) can be almost completely described in a two-dimensional space. We encourage further studies using a smaller number of not redundant orbits that could potentially improve the performance of $GCD$.

## Supporting information

**S1 Fig. Quality of clustering (AUPR) for pairs of models.** Diagram of the matrix of AUPR ($A_{M_1,M_2}$) (a) Erdős-Rényi vs Geometric (b) Erdős-Rényi vs Watts-Strogatz small-world (c) Fitness Scale-Free vs Geometric (d) Geometric vs Watts-Strogatz small-world (e) Fitness Scale-Free vs Watts-Strogatz small-world. For each pair of models, and for each order (from 10 to 100) and edge density (from 0 to 1) combination, the quality of clustering between 100 graphs of each of the two types of models is assessed by the Area Under the Precision-Recall curve (AUPR). A maximum value of 1 corresponds to perfect discrimination. Black crosses represent zero AUPR. (TIF)

**S2 Fig. Probability of correctly distinguishing two random graph models.** with different order and/or edge density. Each block $(i, j)$ concerns the comparison of an $M_1$ of order $N$ and a $M_2$ of order $N'$, with edge density $d$ and $d'$ respectively ranging from 0 to 1 (grey gradient from white to black on the top and right side) with $(M_1, M_2) \in \{ER, SF, SW, GO\}^2$ and $M_1 \neq M_2$. Dashed lines in each block highlight comparison when $d = d'$. (a) Probability matrix $B_{M_1,M_2}$ that $D_{M_1(N,d),M_2(N',d')} > max(D_{M_1(N,d),M_1(N',d')}, D_{M_2(N,d),M_2(N',d')})$ with $M_1 = SF$ and $M_2 = GO$. (b) $B_{ER, SW}$, (c) $B_{SF, GO}$, (d) $B_{GO, SW}$, (e) $B_{SF, SW}$. (TIF)

**S1 Eq. To account for the difference in variability between the correlation coefficients of each pair of orbits, we also computed the following standardised distance $\delta_{std,M_k}$ between $GCM(M_k)$ and $\overline{GCM}_M$ where $\sigma(i, j)$ is the standard deviation of the correlation coefficients of the pair of orbits $(i, j)$ under $H_0$.** We built the test by computing $\eta$ the number of times the standardised distance between $GCM_G$ and $\overline{GCM}_M$ is smaller or equal to the distance $\delta_{std,M_k}$. The $p$-value [50] is defined by $\hat{p} = (\eta + 1)/(K + 1)$. The larger the $p$-value, the less evidence against $H_0$. (PDF)

**S1 Table. Estimated p-values (std).** Each empirical graph is associated with an estimated $p$-value ($\hat{p}$) of being an outcome of an Erdős-Rényi, Fitness scale-free model, a Watts-Strogatz small word or a Geometric model. As in Table 1, empirical graphs are sorted according to their order. ($\hat{p}^* < 0.05$, $\hat{p}^{**} < 0.01$ and $\hat{p}^{***} \leq 0.001$)
(PDF)

## Acknowledgments

We thank Sophie Lanco-Bertrand and Julien Lebranchu (IRD-Sète) for their helpful comments and discussions. The authors would like to thank the SIH (Système d'informations Halieutiques-IFREMER) for the French fleet dataset.

## Author Contributions

**Conceptualization:** Jérôme Roux, Nicolas Bez, Stéphanie Mahévas.

**Formal analysis:** Jérôme Roux.

**Funding acquisition:** Stéphanie Mahévas.

**Methodology:** Jérôme Roux, Nicolas Bez, Paul Rochet, Rocío Joo, Stéphanie Mahévas.

**Project administration:** Nicolas Bez, Stéphanie Mahévas.

**Supervision:** Nicolas Bez, Stéphanie Mahévas.

**Validation:** Jérôme Roux.

**Visualization:** Jérôme Roux.

**Writing – original draft:** Jérôme Roux, Nicolas Bez, Stéphanie Mahévas.

**Writing – review & editing:** Jérôme Roux, Nicolas Bez, Paul Rochet, Rocío Joo, Stéphanie Mahévas.

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
