## [Decision Letter · Decision Letter 0]

18 May 2022

PONE-D-22-11020Graphlet correlation distance to compare small graphsPLOS ONE

Dear Dr. Roux,

Thank you for submitting your manuscript to PLOS ONE. After careful consideration, we feel that it has merit but does not fully meet PLOS ONE’s publication criteria as it currently stands. Therefore, we invite you to submit a revised version of the manuscript that addresses the points raised during the review process.

The originality of your initial idea was noted by reviewers but so was the lack of depth in shaping this idea. Despite the seriousness of such a lack, I would like to give you a chance of refining your work. Please, do carefully consider the reviewers' comments as guidelines for enriching and validating your approach.

We look forward to receiving your revised manuscript.

Kind regards,

Frederique Lisacek

Academic Editor

PLOS ONE

Journal Requirements:

Reviewers' comments:

Reviewer's Responses to Questions

**Comments to the Author**

1. Is the manuscript technically sound, and do the data support the conclusions?

Reviewer #1: Yes

Reviewer #2: Yes

Reviewer #3: Partly

2. Has the statistical analysis been performed appropriately and rigorously? 

Reviewer #1: Yes

Reviewer #2: Yes

Reviewer #3: No

3. Have the authors made all data underlying the findings in their manuscript fully available?

Reviewer #1: Yes

Reviewer #2: Yes

Reviewer #3: Yes

4. Is the manuscript presented in an intelligible fashion and written in standard English?

Reviewer #1: Yes

Reviewer #2: Yes

Reviewer #3: No

5. Review Comments to the Author

Reviewer #1: This is a cute little paper that introduces a simple and elegant method of using the pairwise GCD(M_i,M_j) between all pairs of some set of graphs in order to detect conglomerate “clusters”, if you will, to build statistical tests for classification of graphs into synthetic models or to ask if two empirical graphs come from the same model.

I really like the paper… but right now it’s too “thin”. While the initial *inspiration* is brilliant, the implementation and (lack of) analysis feels like the result of a summer project from an undergrad. While the system is certainly a clever and novel, there’s virtually no *insight* provided. They simply build empirical p-values, AUPRs, and a few simple (though reasonable) statistical tests, and then walk the reader through a bunch of observations of the resulting plots without providing any in-depth analysis or insight that can be gained. There’s so much more low-hanging fruit they could explore without much effort—for example the analytical p-values I mention below… but they could also try to *explain* a bit about the GCM’s and GCD’s among ER graphs, using a few graphlets and orbits as examples.

Basically there’s almost no theoretical development, and it seems to me that it wouldn’t be too difficult to implement some.

I’m going to mark the paper “Major Revisions”, but not because it currently sucks. It doesn’t. The idea is wonderful… you just need to add more substance, more theoretical development (*novel* development), and insight… and I expect it will require significant time and effort to do that.

For Equation (8), my guess is each element (i,j) of the GCM’s will be normally distributed with mean equal to the mean of your \\bar{GMC_M}{i,j}. Why not compute the standard deviations as well, at which point you can compute analytical p-values for each element of a particular GCM_M(i,j)? Then you can take the largest (or smallest) p-value and perform a Bonferroni correction (or use Empirical Brown’s Method) to gather an *analytical* p-value? Then compare the analytical one with the one you use here and explain any differences.

Finally.... PLEASE include better figures in the next round. The resolution is TERRIBLE and the figures are virtually illegible... and not numbered! And please put the figures IN LINE, ABOVE THE CAPTIONS. Putting figures at the end is a 20th-century practice and may be required by the journal once the paper is ACCEPTED... but at the review stage it's a bloody pain in the ass to flip back-and-forth from the back of the paper to the figure caption.

See also attached marked-up PDF for more comments.

Reviewer #2: This paper investigates the effectiveness of the Graphlet Correlation Distance (GCD) for comparison of small graphs. The authors developed a statistical test and designed a comprehensive set of experiments to assess the ability of GCD to distinguish between small graphs generated according to Erdos-Renyi (random graph), Barabasi-Albert (scale free) and k-regular generative models. They apply their findings to the empirical graphs of observed proximities between fishing vessels.

Overall, the paper is well-written and indeed covers the gaps in the literature regarding the applicability of GCD. The evaluation design is comprehensive and supports the authors' conclusions. I only have a few minor comments:

* The authors correctly discuss the fact that for low and high edge densities, especially for small graph orders, ER and BA-SF models cannot be distinguished by GCD nor probably by any other means since such graphs are essentially the same. Perhaps this line could be emphasized somewhat more in the discussion. Is there any theoretical literature covering the question of "when" (in terms of edge density) the graph topology of small graphs becomes truly distinct from random graphs?

* I also think that the authors should have investigated in depth one example where GCD fails to reliably distinguish between

ER or BA-SF generated instances of different edge densities. This could be visualized by embedding GCMs in 2D using say UMAP or tSNE and then further considered by looking at a few particular GCMs and orbits. This would provide a deeper understanding the limitations and strengths of GCD.

* Finally, the authors may want to include the following paper as an example of an application of GCD. A method called correlation graphlet analysis (CGA) is proposed there (details in supplement, code available at https://github.com/astojmir/cgamodules) to select robust correlation thresholds in gene co-expression networks and derive modules from them.

There is also a typo or omission on the lines 169 and 170 - "computational ..." - a word is possibly missing.

Reviewer #3:

━━━━━━━━━━━━━━━━━━━━━━━━━━━━━━━━━━━━━━━━━━━━━━━━━━

PEER REVIEW "GRAPHLET CORRELATION DISTANCE TO

COMPARE SMALL GRAPHS"

━━━━━━━━━━━━━━━━━━━━━━━━━━━━━━━━━━━━━━━━━━━━━━━━━━

Paper summary

The state-of-the-art heuristic to measure the topological similarity between two networks is the graphlet correlations distance (GCD),which is defined as the matrix of pairwise rank correlations between the occurrences of nodes in a network on different graphlets (small induced sub-graphs). Initially suggested for large-scale networks (i.e. networks of more than a 1000 nodes), the authors investigate the applicability of GCD to small networks (i.e., networks of 5 to a 100 nodes). To support their investigation, the authors suggest a method to assess if a given network is statistically significantly similar to a type of model network (i.e., a randomly generated synthetic networks with known topological properties).

Review

══════

To measure the significance of the topological fit of a given network to a model network, the authors design a test to compute an empirical p-value (i.e., how likely is it that I measure a GCD distance of 'x', between the network and generated model networks by chance). Although methodologically correct, computing an empirical p-value is quite trivial and hence the methodological contribution of this paper is very minor. All the more as a Mann-Whitney-U test has already been designed and applied for the same thing (see [1]).

The presented analysis is incomplete. Firstly, although GCD is the state-of-the-art distance measure for large networks, it has not been

shown to be so for small graphs (5-100 nodes), so a comparison w.r.t. other methods should be included. All the more so, as the statistical power of GCD for such small graphs is questionable. This, as GCD is based on correlations between graphlet counts, and computing the correlations over fewer than 20 data points (nodes) is unlikely to capture much signal. Lastly, the authors only consider three types of model networks in their analysis, which is very few considering the plethora of model networks available. For instance, the paper introducing GCD considered seven model networks.

The quality of writing in this paper is not up to a scientific standard, as many concepts are either vaguely defined, not defined or defined after they are first used. For instance, graphlets in this paper are defined as 'small connected sub-graphs that extend the concept of motifs'. Not only is this definition of graphlets wrong, graphlets are 'small connected non-isomorphic and induced subgraphs' [2], motifs have also not been defined in this paper, so the comparison is uninformative. Also, without an intuitive explanation, a non-expert reader will not understand what a graphlet is from this formal definition.

Conclusion

══════════

Given the small methodological contribution, the incomplete analysis and the quality of the writing, I would recommend to reject this paper.

References

══════════

[1] Windels, S. F., Malod-Dognin, N., & Pržulj, N. (2019). Graphlet Laplacians for topology-function and topology-disease relationships. Bioinformatics, 35(24), 5226-5234.

[2] Pržulj, N., Corneil, D. G., & Jurisica, I. (2006). Efficient estimation of graphlet frequency distributions in protein–protein interaction networks. Bioinformatics, 22(8), 974-980.

6. PLOS authors have the option to publish the peer review history of their article (what does this mean?). If published, this will include your full peer review and any attached files.

Reviewer #1: No

Reviewer #2: No

Reviewer #3: No

---

## [Decision Letter · Decision Letter 1]

25 Oct 2022

PONE-D-22-11020R1Graphlet correlation distance to compare small graphsPLOS ONE

Dear Dr. Roux,

Thank you for submitting your manuscript to PLOS ONE. After careful consideration, we feel that it has merit but does not fully meet PLOS ONE’s publication criteria as it currently stands. Therefore, we invite you to submit a revised version of the manuscript that addresses the points raised during the review process. The contrasted comments of the original two reviewers led me to ask for a third opinion and the concerns about your definition of domain of applicability, the appropriateness of GCD and the conclusions you draw from a rather limited case study are shared by the new reviewer. It is essential that you properly address these concerns in a revised version.

We look forward to receiving your revised manuscript.

Kind regards,

Frederique Lisacek

Academic Editor

PLOS ONE

Reviewers' comments:

Reviewer's Responses to Questions

**Comments to the Author**

1. If the authors have adequately addressed your comments raised in a previous round of review and you feel that this manuscript is now acceptable for publication, you may indicate that here to bypass the “Comments to the Author” section, enter your conflict of interest statement in the “Confidential to Editor” section, and submit your "Accept" recommendation.

Reviewer #2: All comments have been addressed

Reviewer #3: (No Response)

Reviewer #4: (No Response)

2. Is the manuscript technically sound, and do the data support the conclusions?

Reviewer #2: Yes

Reviewer #3: No

Reviewer #4: No

3. Has the statistical analysis been performed appropriately and rigorously? 

Reviewer #2: Yes

Reviewer #3: No

Reviewer #4: No

4. Have the authors made all data underlying the findings in their manuscript fully available?

Reviewer #2: Yes

Reviewer #3: Yes

Reviewer #4: Yes

5. Is the manuscript presented in an intelligible fashion and written in standard English?

Reviewer #2: Yes

Reviewer #3: Yes

Reviewer #4: No

6. Review Comments to the Author

Reviewer #2: In this revision, the authors added an investigation about the variability of individual Graphlet correlation coefficients with respect to change in density and order. This view could potentially form a starting point for designing an improved distance measure between graphs, perhaps by incorporating terms related to degree or density.

This version is definitely an improvement to the previous version, addressing many of the reviewers’ comments, but could probably go a bit further to understand the issues associated with using GCD (or any other graph topology similarity measure) for smaller graphs.

Overall, I recommend acceptance, but I hope that in the future we would see a somewhat deeper treatment of this topic.

Reviewer #3: ━━━━━━━━━━━━━━━━━━━━━━━━━━━━━━━━━━━━━━━━━━━━━━━━━━

PEER REVIEW REVISION 1: "GRAPHLET CORRELATION

DISTANCE TO COMPARE SMALL GRAPHS"

━━━━━━━━━━━━━━━━━━━━━━━━━━━━━━━━━━━━━━━━━━━━━━━━━━

Main issues.

════════════

• The main conclusion of this paper is that 2 fishing fleets are of

ER topology. To me, this conclusion could have been achieved by looking at the

degree distributions and fitting them with a poisson distribution.

• The 'domain of applicability', defines combinations of graph sizes

(in terms of number of nodes) and network densities where GCD can be used

to distinguish networks of different topologies. This domain is

determined for graphs with known and very specific topologies (ER

graphs, SF graphs and k-regular graphs). I wonder how valid this

boundary of applicability is for the comparison to the real world

networks (fishing fleet networks), which are of unknown topology. In

my opinion, the "boundary of applicability" is only applicable in a

generic fashion if the authors were to consider a more varied range of

model networks.

• From Figure 5, fleet 1 has a clear 1-regular structure. As no other

orbit but orbit 0 (an edge) is ever touched, I would argue that the

GCM is undetermined (a correlation between different orbit counts

can not be determined if all orbit counts are simply 0). So, to me,

Figure 5B is wrong, as the GCM for such a network is not an 11 by 11

matrix of 1's but one of NAN's. Secondly, GCD is a heuristic to

summarise the topology of a network. There simply is no topology to

summarise in fleet 1, so I don't see why fleet 1 is even included in

this analysis. Thirdly, the authors mention that according to their

GCD basted tests, the networks in fleet 1 are not of ER

topology. However, as the GCM's are quite similar to those of ER

networks due to some graphlets not occurring, they argue, the

networks in fleet 1 are in fact to some extent like ER

networks. This is false, apart from orbit 14, all orbits occur in an

ER graph, the networks in this analysis are just too small for them

to do so. Again, an argument for why GCD simply is not

applicable. Also, a network can not be of 1-regular topology and of

ER topology, so the text is contradictory.

• As is clear from my previous comments, I don't think GCD is the

correct tool here. It would help if the authors showed it actually

performed better than other distance metrics (like simply the

distance between degree distributions).

False statements

════════════════

• "The shift to quantitative graph comparisons with the introduction

of similarity or distance measures is more recent[23]".

The paper referenced dates from 2014. Network distance measures have

been introduced long before that. Even graphlet based network

comparison was introduced before in 2004.

Sentences containing typos and/or grammatical mistakes

══════════════════════════════════════════════════════

• This comparison is often done in a descriptive and qualitative way

by comparing synthetic indicators of graph topology i.e the

configuration by which the individuals of a graph are connected.

i.e. is written two dots.

• In fact, in many graphs, the node degree distribution seems to

follow a power law whose power γ is comprised between 2 and 3.

Comprised is not the right word here.

• On one hand, the increase in order allows the emergence of complex

topologies consistent with the topological properties of the model ( [52]);

on the other hand, the Spearman’s Correlation coefficient becomes more accurate

when computed on a larger number of nodes ([53]); …

Why the brackets around the references?

Vague definitions

═════════════════

While I agree with the authors that technical definitions can scare a lay audience,

when intuitions are provided instead they should at least be correct and complete.

• Graphlets are small connected non-isomorphic (different number of

nodes and/or connected in a different way [28])

I don't think anyone understands what non-isomorphic means in this

context from this definition. Two nodes are isomorphic if, when you

swap their node label (position in the network), you still end up with

the same network/graphlet.

• The authors mention that graphlets are an extension to motifs. How

so? Graphlets are induced, motifs are not. The way it is written it

just looks as if the authors don't know.

Reviewer #4: In this paper, the authors aim to investigate the performance of the Graphlet Correlation Distance (GCD) on graphs with small numbers of nodes and density varying from low to high. GCD is a distance measure between graphs based on the graphlet degree distributions, which enumerate in a graph how many times each node appears on each orbit within a chosen set of graphlets. Graphlet degrees are a generalisation of the concept of node degree (degree = single-link graphlet orbit count). The aim is to establish a domain of applicability for small graphs within which graph comparison using GCD could be meaningful.

However, the paper falls short of this goal in many ways. The "domain of applicability" with respect to number of nodes and graph density is established based on separability of Erdős-Rényi (ER) and Barabási-Albert (BA) random graph models, which is a very narrow application. Furthermore, small BA graphs make little sense; their key feature of power-law degree distribution is hardly apparent with graphs of 5-50 nodes, and the effect of the initial network on the result is non-negligible (what initial graph is used is not explained). Thus, the comparison of BA to ER doesn't make sense to begin with, and even if it did, their separability is not indicative of any general "domain of applicability" that could be used to inform whether GCD is applicable to any other separation or clustering task of arbitrary small graphs.

The paper proposes a statistical test to tell if a graph is similar to the ER or BA model. This is a simple simulated sampling p-value calculation based on simulations of the models, using GCD as the test statistic. Using GCD for this task makes little sense, especially for comparison with the ER model, where edges are independent. Since the "calculate p-value through simulations" process is elementary textbook material and the simulation results here are restricted to ER and BA models, there is no wider applicability to the statistical testing section.

The empirical graphs section details two classes of fishing vessel graphs, the first of which consists of 1-regular graphs. For a 1-regular graph, the first graphlet degree (i.e. node degree) is 1 and every other graphlet degree is 0 for every node. It does not make any sense to use any graphlet-based methods on these graphlet degree distributions. All the graphs have identical highly degenerate graphlet degree counts. It makes absolutely zero sense to use GCD with these graphs. Earlier in the paper, 1-regular graphs are compared to simulated ER and BA with GCD as well, which is also nonsensical. Including nonsense results is unnecessary and they should be omitted.

With the second class of empirical fishing vessels, it is concluded that they are in the "domain of applicability". What is meant is that they are in the domain where ER and BA models can somewhat be separated from each other. This does not translate to applicability to any arbitrary graphs, such as these fishing vessel graphs.

Throughout the paper, GCD-11 is used. This measure takes into account 11 specific orbits, but is not the only GCD measure possible. Any set of graphlets and their orbits can be used to construct a GCD. The analysis is therefore very limited. It would be better to look more at individual graphlet degrees and different subsets of graphlets chosen to be included in the GCD, and how all the different graphlet degrees vary in small graphs. More graph structures should be considered, not only ER and BA.

In the Conclusion section, it is stated that "However, due to the behaviour of the Graphlet [sic] correlation coefficients in response to change in density, we expect the domain of applicability between the Erdős-Rényi and Barbási-Albert [sic] scale-free graph models described in this work to be qualitatively generic to other pair [sic] of models allowing a density ranging from 0 to 1." There is nothing in the paper to back up this claim and I would be highly surprised if it was true. Setting up a general domain of applicability for GCD in small graphs is the goal of this paper and the main claim regarding that is unsubstantiated.

The idea of the paper is solid, but the analysis is too limited to provide any meaningful insight.

Regarding the fulfilment of the 7 criteria that PLOS One has for publication, I evaluate the current paper as follows:

1. The study presents the results of original research.

Fulfilled to my knowledge.

2. Results reported have not been published elsewhere.

Fulfilled to my knowledge.

3. Experiments, statistics, and other analyses are performed to a high technical standard and are described in sufficient detail.

Not fulfilled. The premise of the paper is interesting but all the actual content is based on an extremely limited view of "separation task between ER and BA models" and therefore does not meet high technical standards because of glaring shortcomings and limitations.

Everything in the paper concerning 1-regular graphs (simulated or empirical) is nonsense and should be omitted.

4. Conclusions are presented in an appropriate fashion and are supported by the data.

Not fulfilled. The conclusions are based on the very limited analysis and are not supported in any general sense.

5. The article is presented in an intelligible fashion and is written in standard English.

Mostly fulfilled. Some clear language deficiencies remain.

6. The research meets all applicable standards for the ethics of experimentation and research integrity.

Fulfilled to my knowledge.

7. The article adheres to appropriate reporting guidelines and community standards for data availability.

Fulfilled to my knowledge.

7. PLOS authors have the option to publish the peer review history of their article (what does this mean?). If published, this will include your full peer review and any attached files.

Reviewer #2: No

Reviewer #3: No

Reviewer #4: No

---

## [Author Response · Author response to Decision Letter 1]

9 Dec 2022

Please find our responses to the reviewers as a pdf file in the "Attach Files" section.

---

## [Decision Letter · Decision Letter 2]

4 Jan 2023

PONE-D-22-11020R2Graphlet correlation distance to compare small graphsPLOS ONE

Dear Dr. Roux,

Thank you for submitting your manuscript to PLOS ONE. After careful consideration, we feel that it still lacks a few details to fully meet PLOS ONE’s publication criteria as it currently stands. Therefore, we invite you to submit a revised version of the manuscript that addresses the points raised during the review process. The missing details are sparse. The reviewer accurately pointed a few minor issues to attend to in order to improve the reading flow and mostly requested the clarification of the hidden meaning of "ensuring relevant computations".

We look forward to receiving your revised manuscript.

Kind regards,

Frederique Lisacek

Academic Editor

PLOS ONE

Journal Requirements:

Reviewers' comments:

Reviewer's Responses to Questions

**Comments to the Author**

1. If the authors have adequately addressed your comments raised in a previous round of review and you feel that this manuscript is now acceptable for publication, you may indicate that here to bypass the “Comments to the Author” section, enter your conflict of interest statement in the “Confidential to Editor” section, and submit your "Accept" recommendation.

Reviewer #4: (No Response)

2. Is the manuscript technically sound, and do the data support the conclusions?

Reviewer #4: Yes

3. Has the statistical analysis been performed appropriately and rigorously? 

Reviewer #4: Yes

4. Have the authors made all data underlying the findings in their manuscript fully available?

Reviewer #4: Yes

5. Is the manuscript presented in an intelligible fashion and written in standard English?

Reviewer #4: Yes

6. Review Comments to the Author

Reviewer #4: Revision 2 is much, much better than revision 1. The analysis has been extended to more network models, and the text has been revised to more accurately reflect the content of the work. Revision 2 is in a state that it fulfills, to my knowledge, the criteria for publication in PLOS ONE.

Having said that, I have some minor comments that would be good to be addressed in my opinion, for the benefit of the reader. One of them concerns the methodology (more important), and the rest are mostly about the presentation and wording (less important). If these are fixed, I can recommend publishing.

Methodology:

- Line 157-158: diagonals of D_M2,M1 and D_M1,M2 are removed, but I don't see why. The first element of the diagonal of D_M1,M2 is the distance between model 1 graph 1 and model 2 graph 1, which are different graphs, and I don't see why their distance should be excluded. I don't understand the reason that is given ("Ensuring relevant computations"), so it would be good to give a valid reason as to why the diagonals have been removed.

Other:

- When talking about the order of a graph for the first time, it would be good to define it as the number of nodes, since "number of nodes" is more common terminology than "order" and some readers are probably unfamiliar with the latter (the first mention is in the abstract).

- From line 138 onwards: the sentence "For a given order N in {10,11,...,50,55,...,100} with card{N} = |N| = 51, and a given edge density d in {0,0.01,...,1} with card{d} = |d| = 101 ..." has some issues. First, the range of N should be more clearly presented, such that the steps between consecutive values and the transition point are unambiguously clear (for example "N in {10,11,...,48,49,50,55,60,65,...,100}"). Secondly, N is the number of nodes (an integer, a member of the set {10,11,... and so on}), so notation card{N} isn't clear (a single integer is not a set) and notation |N| also seems to refer to the absolute value of the number of nodes. What is actually meant seems to be that there were 51 different values for N (i.e. the size of {10,11,...,48,49,50,55,60,65,...,100}). This should be just written out instead of using tricky notation (incorrectly).

- Equations 11 and 12: these are equations for graphlet correlation distance (GCD) between the mean GCM_M and GCM_Mk or GCM_G. It would be helpful to write that in the text, so the reader is aware that this equation actually computes GCD.

- Line 182: "A probability of 1 ...", does this sentence refer to the percentage explained in the previous sentence or to the probability matrix B (defined later)? It is confusing to switch terminology between "percentage" and "probability".

- Line 256: "the surface of the domain of separability" has no defintion but seems to refer to the proportion of matrix elements where the value is greater than 0.9 (based on caption of fig. 3). It would be good to explicitly define what "surface of the domain of separability" means here, where the term first appears.

- The final paragraph of the Conclusion section, from line 393 onwards, is worded strangely, and I'm not quite sure what it's trying to say. It would be good to clarify the language.

7. PLOS authors have the option to publish the peer review history of their article (what does this mean?). If published, this will include your full peer review and any attached files.

Reviewer #4: No

---

## [Editor Report · Decision Letter 3]

30 Jan 2023

Graphlet correlation distance to compare small graphs

PONE-D-22-11020R3

Dear Dr. Roux,

We’re pleased to inform you that your manuscript has been judged scientifically suitable for publication and will be formally accepted for publication once it meets all outstanding technical requirements.

Kind regards,

Frederique Lisacek

Academic Editor

PLOS ONE
---

## [Editor Report · Acceptance letter]

3 Feb 2023

PONE-D-22-11020R3 

Graphlet correlation distance to compare small graphs 

Dear Dr. Roux:

I'm pleased to inform you that your manuscript has been deemed suitable for publication in PLOS ONE. Congratulations! Your manuscript is now with our production department. 

Kind regards, 

on behalf of

Dr. Frederique Lisacek 

Academic Editor

PLOS ONE